# An Open Quantum Chemistry Property Database of 120 Kilo Molecules with 20 Million Conformers

## Abstract

Artificial intelligence is revolutionizing computational chemistry, bringing unprecedented innovation and efficiency to the field. To further advance research and expedite progress, we introduce the Quantum Open Organic Molecular (QO2Mol) database a large-scale quantum chemistry dataset designed for researches on organic molecules under an open-source license. The database comprises 120,000 organic molecules and more than 20 million conformers, encompassing 10 different elements (C, H, O, N, S, P, F, Cl, Br, I), with heavy atom counts exceeding 40. Each conformation was computed at B3LYP/def2-SVP level of theory to derive quantum mechanical properties, including potential energy and forces. The molecules included in the dataset are based on fragments from compounds in ChEMBL, ensuring their structural *relevance to real-world compounds*. The extensive variety of molecular structures and elemental compositions represented in the dataset can facilitate construction of potential energy surface and various downstream tasks.

## 1 Introduction

The advent of artificial intelligence (AI) has heralded a new era of innovation and efficiency in computational chemistry. Among the various areas of focus within computational chemistry, the study of small organic molecules holds a particularly prominent position due to their fundamental importance in diverse scientific disciplines, including drug discovery (Mayr et al., 2016; Chen et al., 2023; Agüero-Chapin et al., 2022; Stokes et al., 2020; Zeng et al., 2022), reaction prediction (Żurański et al., 2021; Wang et al., 2023; Pereira & Trofymchuk, 2023; Lin et al., 2023; Ding et al., 2024), and materials science (Yang et al., 2020; Cheng et al., 2021; Dai et al., 2021; Bu et al., 2022; 2023).

However, there is currently still a shortage of publicly available large-scale quantum chemistry datasets to support the increasingly extensive research on small organic molecules by AI and computational chemistry experts in the field. Existing public quantum chemistry datasets are either constrained by limited elemental diversity and molecular variety, or by a small sample size predominantly focused on small molecules with low heavy atom counts, thereby lacking the necessary breadth and comprehensiveness for robust research applications. Figure 1 illustrates that other commonly used datasets are restricted in both their coverage of element types and the number of conformers they encompass. We provide a more detailed comparison and description of the shortcomings of existing datasets in Section 3.2.

To address these challenges and to promote deeper development in the field, we release Quantum Open Organic Molecular (QO2Mol) database, the large-scale quantum chemistry dataset with 20 million conformers, designed for the research in molecular sciences under an open-source license. We provide a comprehensive set of molecular property labels, encompassing potential energy, forces, and formal charge, and additional relevant attributes. In Figure 1, compared to other well-known datasets, QO2Mol covers the widest variety of 10 elements and includes the largest number of conformers. Additionally, QO2Mol employs high-precision quantum mechanical calculations, which are computationally intensive and costly. Refer to Section 4.5 for computation costs. By offering this high-quality data to the global scientific community, we aim to accelerate advancements in com-

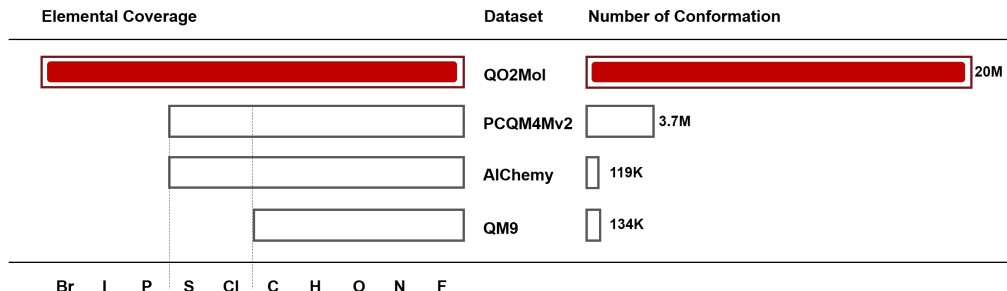

Figure 1: Main characteristics of commonly used datasets regarding elemental coverage and the number of molecular structures. The left panel illustrates the coverage of elements; The right panel presents the number of conformations.

putational chemistry, material science, and drug discovery. In summary, our key contributions are threefold:

- Firstly, we introduce the QO2Mol dataset, which comprises 120,000 organic molecules and more than 20 million conformations. This database covers 10 different elements with heavy atom counts exceeding 40, closely mirroring the distribution of chemical structures found in widely used real compound libraries.

- Secondly, we employ B3LYP/def2-SVP level of theory and basis set to obtain reliable molecular property labels, including potential energy and forces, providing a valuable database for future research and model development.

- Finally, we provide scripts for loading and processing the dataset, along with benchmark code and comparative results, enabling researchers to quickly get started and easily integrate the dataset into their projects. All scripts and codes are available at `https://github.com/saiscn/QO2Mol/`.

We hope these contributions would effectively advance the field of computational chemistry and provide essential resources and methodologies for accurate molecular modeling.

## 2 BACKGROUND INFORMATION

### 2.1 POTENTIAL APPLICATIONS OF OUR DATASET ON DATA-DRIVEN METHODS

This section explores the potential impacts of our dataset on three specific areas: Potential Energy Surfaces, Force Field Models, and Conformation Generation. However, it is crucial to recognize that the scope of influence may reach well beyond these identified domains.

**Potential Energy Surface** The potential energy surface (PES) of atomistic systems is the core of several aspects of physical chemistry, such as transition states, vibrational frequencies and electronic properties. Many of current methods based on deep learning mechanism focus on deploying neural networks to predict QM computed properties (Qiao et al., 2020; Atz et al., 2021; Walters & Barzilay, 2021; Chen et al., 2021; Wang et al., 2022). These methods directly predict the QM properties instead of solving the many-body Schrodinger equation numerically. All these methods require high-precision QM data for training reliable models, which is what the QO2Mol dataset can provide.

**Force Field Models** Force fields are typically employed in downstream tasks like molecular dynamics simulations and structure optimizations (Joshi & Deshmukh, 2021; Shub et al., 2013; Suzuki et al., 2022; Souza et al., 2021; Bejagam et al., 2020). They are essential for understanding and predicting the behavior and properties of molecular systems. Our dataset encompasses a diverse range of element types and molecular structures, which is valuable for fitting or validating high-accuracy force field models.

**Conformation Generation** The molecular conformation generation task aims to quickly obtain reasonable and stable atomic 3D coordinates, which can be used for downstream tasks such as molecular property prediction and molecular docking. Traditional methods acquire reliable 3D structures through DFT calculations, but the computational costs become increasingly expensive as the number of atoms increase. Recently, many studies have utilized neural network models to directly generate conformations from molecular graphs (Simm & Hernandez-Lobato, 2020; Shi et al., 2021; Zhu et al., 2022). Our dataset includes rotational scans of all flexible bonds for each flexible molecule and can serve as a training set for the molecular conformation generation task.

## 2.2 BASIC CONCEPTS OF COMPUTATIONAL CHEMISTRY

We introduce the necessary preliminaries of computational chemistry that will be used later.

- Density Functional Theory (DFT) (Thomas, 1927) is a popular computational method used to approximately solve Schrödinger equation of molecular systems which offers energies labels and possibly estimate further molecular property labels from the computed solution.

- Force fields can be applied in various areas of computational chemistry, such as Free Energy Perturbation (FEP) calculations (Jiang & Roux, 2010; Wang et al., 2015).

- InChI (Heller et al., 2015) (The International Chemical Identifier) is a unique representation of a chemical substance. InChI decomposes molecular graphs into a series of layered descriptive information, accurately capturing the chemical structure of the molecule. InChIKey (Pletnev et al., 2012) is a compacted version of InChI with 27-character fixed-length. InChIKey is intended for identifying a unique molecule in database searching/indexing (Wikipedia contributors, 2024).

- SMILES (Weininger, 1988) (Simplified Molecular Input Line Entry System) is a ASCII string that represents a chemical structure in a way that can be friendly used by the computer. It encodes molecular graph notations into compact linear strings through Depth First Search (DFS) algorithm.

- Heavy atom is any atom other than hydrogen, typically used in molecular studies to focus on more complex atomic interactions. Heavy atoms form the structural backbone of molecules, defining their geometry and functional groups, while hydrogen atoms are typically peripheral and less influential in determining molecular properties. Thus distinguishing heavy atoms is highly relevant to real-world applications across various fields.

## 2.3 CALCULATION PRECISION

In quantum chemistry, computational precision is closely tied to the choice of calculation methods and basis sets. Advanced methods offer higher precision but demand substantial computational resources. Among DFT calculation functionals, B3LYP (Becke, 1988; Lee et al., 1988; Becke, 1993; Stephens et al., 1994) is the one of most popular choices in quantum mechanical calculations of organic molecular systems due to its balance between computational efficiency and precision. In QO2Mol, we employ the B3LYP/def2-SVP level of theory, one of the highest precision levels achievable within an acceptable computational cost range for large-scale calculations of organic molecular systems.

Table 1: Summary of main characteristics of the molecules in QO2Mol dataset.

| Property | Mean | Std | Max |
|---|---|---|---|
| Number of atoms | 23.68 | 8.40 | 111 |
| Number of heavy atoms | 12.62 | 4.49 | 53 |
| Molecular weight (amu) | 186.27 | 65.51 | 1139.76 |
| Number of rotatable bonds | 2.03 | 1.51 | 26 |
| Number of rings | 1.50 | 0.92 | 19 |
| Number of conformations | 204.71 | 307.29 | 11950 |

# 3  QO2MOL AND PREVIOUS DATASETS

## 3.1  OVERVIEW OF QO2MOL DATASET

Overall, QO2Mol dataset encompasses 120 kilo molecules, with 10 elements (H, C, N, O, F, P, S, Cl, Br, and I). Each structure employs calculation with B3LYP/def2-SVP level of theory. We provide statistics for the main characteristics of the molecules in QO2Mol dataset in Table 1. QO2Mol is primarily composed of small organic molecules with an average of about 12 heavy atoms, featuring up to 19 rings and 26 rotatable bonds. The average molecular weight is approximately 186.27. Each molecule has averagely 204 conformations. The smallest molecule in QO2Mol is methane, which has only one conformation. The largest molecule contains 111 atoms, including 53 heavy atoms and 13 rings.

Figure 2: Illustration of the maximum molecule in QO2Mol with 111 atoms and 53 heavy atoms. (a) The left has 35 conformations. (b) The right has 74 conformations in the dataset.

## 3.2  COMPARISION WITH PREVIOUS DATASETS

Table 2: Summary of main characteristics among commonly used QM datasets.

| Dataset | Elements | Molecules | Structures | Conformer Task | Heavy Atoms | Method | Year |
|---|---|---|---|---|---|---|---|
| QM9 (Ramakrishnan et al., 2014) | H,C,N,O,F | 134K | 134K | ✗ | 9 | B3LYP/6-31G(2df,p) | 2014 |
| AN1-1 (Smith et al., 2017) | H,C,N,O | 57K | 22M | ✓ | 8 | $\omega$B97x/631G(d) | 2017 |
| AlChemy (Chen et al., 2019) | H,C,N,O,F,S,Cl | 119K | 119K | ✗ | 14 | B3LYP/6-31G(2df,p) | 2019 |
| PCQM4Mv2 (Hu et al., 2021) | H,C,N,O,F,S,Cl | 3.7M | 3.7M | ✗ | 51 | B3LYP/6-31G(d) | 2021 |
| $\nabla^2$DFT (Khrabrov et al., 2024) | H,C,N,O,F,Cl,Br | 1.9M | 15M | ✓ | 27 | $\omega$B97x-D/def2-SVP | 2024 |
| **QO2Mol** | H,C,N,O,F,P,S,Cl,Br,I | 120K | 20M | ✓ | 44 | B3LYP/def2-SVP | 2024 |

We provides a comparative overview of several commonly used quantum mechanical datasets in Table 2, highlighting their respective methodologies, molecular coverage, and elemental diversity. QM9 (Ramakrishnan et al., 2014), employing the B3LYP/6-31G(2df,p) method, contains 134,000 molecules with a maximum of 9 heavy atoms, limited to the elements H, C, N, O, and F. The AN1-1 dataset (Smith et al., 2017), released in 2017, using the $\omega$B97x/6-31G(d) method, features 22 million molecules but is restricted to only 8 heavy atoms and 4 elements (H, C, N, O). Alchemy (Chen et al., 2019), released in 2019, also uses the B3LYP/6-31G(2df,p) method but includes 119,000 molecules, expanding the elemental range to H, C, N, O, F, S, and Cl, and accommodating up to 14 heavy atoms. PCQM4Mv2 (Hu et al., 2021), utilizing data from the PubChemQC Project (Nakata & Shimazaki, 2017) which employs the B3LYP/6-31G(d) level of precision, comprises 3.7 million molecules and includes 10 elements H, C, N, O, F, S, Cl.

Overall, the QO2Mol dataset encompasses the widest variety of elements. Most earlier released datasets like QM9 are severely limited in the number of molecular structures, making them grossly inadequate for training large-scale models. Furthermore, although ANI-1 boasts a considerable sample size, its restriction to only 4 elements (H, C, N, O) imposes a limitation for studying small organic molecules with diverse spectral properties. In addition, PCQM4v2 only provides HOMO-LUMO gap labels, which are insufficient for supporting more complex molecular tasks and studies. The $\nabla^2$DFT dataset, while encompassing a broader range of molecules, has fewer average conformations per molecule compared to QO2Mol. $\nabla^2$DFT focuses more on the diversity of molecules, whereas QO2Mol emphasizes the sampling density of the potential energy surface.

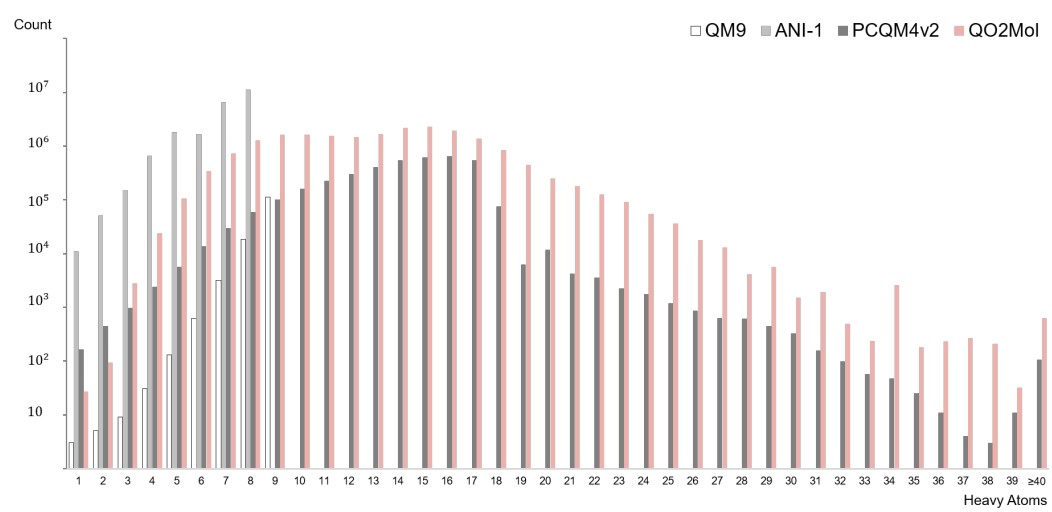

Figure 3: Distribution of the number of conformations with different heavy atom counts among commonly used datasets. We omitted Alchemy because of its small scale.

In Figure 3, QO2Mol exhibits a broad distribution of heavy atom counts and the richest number of conformations overall. In contrast, while ANI-1 offers a substantial number of conformations for smaller heavy atom counts, its limitation to a maximum of 8 heavy atoms severely impacts the diversity and realism of the structures it covers. For example, organic molecular structures with high occurrence rates such as naphthalene (10 heavy atoms) and biphenyl (12 heavy atoms) cannot be incorporated. QO2Mol's extensive molecular and elemental coverage, combined with advanced computational methodology, underscores its superior capacity for quantum mechanical studies, particularly for larger organic molecules and a broader spectrum of elements.

**Remark** We also acknowledge the existence of several other notable datasets in the field, such as OC20/22 (Chanussot et al., 2021; Tran et al., 2023), which is frequently used for crystalline material tasks, and GEOM (Axelrod & Gómez-Bombarelli, 2022). However, these datasets focus on different domains and are not directly designed for the study of small organic molecules. Our dataset specifically addresses the unique challenges and requirements of high-precision quantum mechanical calculations for organic molecules, filling a gap that existing datasets do not cover. This distinction ensures that our contributions are both complementary to and distinct from the current resources available in the field.

## 4 DATASET GENERATION

In this section, we outline the process of data selection, processing, and preparation in QO2Mol. To ensure the quality and reliablity of quantum mechanical data, the following considerations need to be taken into account :

- The selected molecules should represent a chemical space that closely aligns with the distribution of chemical structures found in widely used compound library, such as ZINC (Irwin et al., 2020), PubChem (Wang et al., 2009), and ChEMBL (Gaulton et al., 2012).

- Identify as many key conformations as possible on the potential energy surface, as these play a critical role in determining the properties of the molecules.

- Calculate properties using high-level quantum mechanical methods to ensure accuracy and reliability.

By adhering to these guidelines, we release the QO2Mol dataset, which comprises 120,000 organic molecules and their corresponding 20 million conformations.

### 4.1 MOLECULE FRAGMENTATION

We first derive a set of source compounds from ChEMBL, a widely used virtual screening compound database for drug design (Sadybekov & Katritch, 2023). Performing quantum mechanical calculations directly on these compounds is quite challenging due to the large size of these molecules. To overcome the computational difficulties of quantum mechanical calculations, we employed a Compound Fragmentation Process, dividing the source compounds into smaller fragments containing fewer heavy atoms, as shown in Figure 4. In this way, we ensured that the basic fragment structures can be found in real-world molecules and are therefore chemically meaningful. Then a total of 120,000 fragmented molecules were selected based on three rules: 1) with top 90% occurrence frequency over the database; 2) labeled as important phosphate groups by our chemistry expert; 3) encompassing 10 different elements(C, H, O, N, S, P, F, Cl, Br, I). We also ensured that there was no fragment duplication during the generation procedure by utilizing InChIKey and canonical SMILES identifiers.

Our selection criteria did not impose restrictions on the number of heavy atoms. This approach enables us to capture a diverse range of significant and complex chemical space that might not be adequately represented in existing databases, such as QM9 and ANI-1.

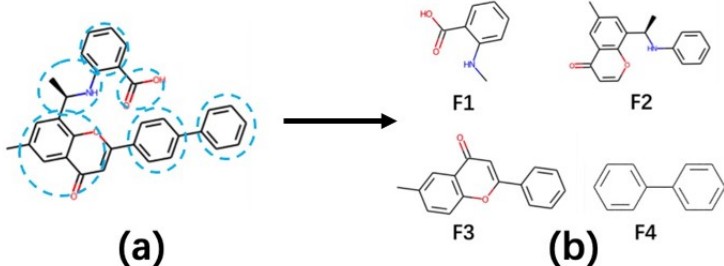

Figure 4: An example of molecule fragmentation process. The molecule (a) is decomposed into four fragments: F1, F2, F3, and F4r.

### 4.2 CONFORMATION GENERATION

The constituent atoms of a molecule exhibit dynamic motion in three-dimensional space, generating the molecule's conformational space. Each conformation has its own unique energy, collectively forming the molecular potential energy surface in 3N-dimensional space. The macroscopic properties of a molecule are effectively described by the ensemble average of the various conformational properties existing on this PES. Thus, the contributions of key conformations, such as local minima or transition state structures, are considerably important, while the significance of other conformations is also noteworthy. Given that, we sampled multiple conformations for each molecule within the QO2Mol database.

**Structure Optimization** For each selected fragment molecule, an initial 3D structure is generated using the RDKit package (Landrum et al., 2013) based on its SMILES (Weininger, 1988) representation. Then each initial structure is optimized to a local minimum at the B3LYP/def2-SVP precision level. To ensure the structure reliability, during the structure optimization process, we employ four convergence criteria to ensure the resulting structures are reasonable: 1) Maximum force <0.00045; 2) root-mean-square force <0.00030; 3) maximum displacement <0.00180; 4) root-mean-square displacement <0.00120. Following each structural optimization, we perform a validation step to ensure that all bond lengths fall within a defined range relative to their empirical values. For example, the empirical length of CC single bond is approximately 1.54 Å as widely observed (Allen et al., 1987). We provide a statistic distribution of C-C bond length over the whole dataset in Figure 5.

**Conformation Search** Conformation search is performed on optimized structures obtained in the previous step. At room temperature, the flexible dihedral angles of molecules are likely to rotate. Therefore, rotation is the most influential factor in constructing potential energy surfaces. Based on this intuition, we perform rotational search in 30-degree increments each step on all rotatable bonds of each molecule. By systematically rotating the flexible bonds of molecule to specific degrees,

a series of new structures are generated. These structures are then optimized at the B3LYP/def2-SVP level with fixed torsions. Additionally, for specific molecules, we also perform stretching and bending operations on bond lengths and bond angles, generating corresponding conformations. We ensure that all bond types, such as C=C and C=O, are included in these manipulations. Moreover, the database includes a collection of nearby unstable conformations for each stable conformation, further enhancing the representation of the overall molecular potential energy landscape. We provide a scan curve showing the potential energy changes during the flexible bond rotation in Figure 5.

Based on the mentioned conformation generation procedure, we finally obtained a total of 20 million conformers for the 120,000 molecules in our database.

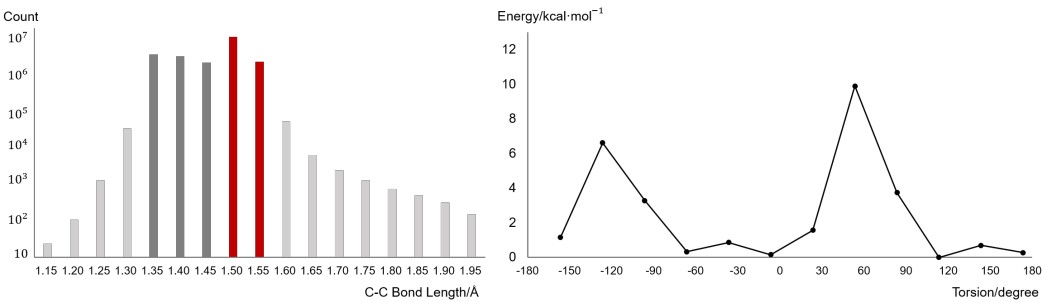

Figure 5: Results of data generation. (left) The distribution statistics of C-C single bond lengths in the dataset. (right) An example of the rotational scan curve. We conduct rotational scan on all flexible bonds of each molecule during conformation search procedure.

### 4.3 PROPERTIES

All conformations were analyzed to compute energy and forces at the B3LYP/def2-SVP level of theory. The forces, representing the first-order derivatives of energy with respect to coordinates, were calculated for each atom in the three Cartesian directions (x, y, z). Among the 20 million conformations, we also provide additional properties for approximately 210,000 stable conformations, although this is not the main focus of our contribution. For these stable conformations, we conducted frequency and charge population calculations. Vibrational frequencies were derived through diagonalization of the Hessian matrix, yielding 3N - 6 frequency values after excluding the three translational and three rotational modes. The Hessian matrix represents the second-order derivatives of energy with respect to coordinates. These frequency calculations allow for the determination of thermodynamic properties, including zero-point energy, entropy, enthalpy, heat capacity, and free energy, utilizing both harmonic and ideal gas approximations. The charge population analysis includes the calculation of electron density-derived charges such as ESP (Electrostatic Potential) charges and Mülliken charges. More details are provided in Appendix B.

### 4.4 DATA SEGMENTATION

In order to support various learning tasks in this field, we divided the data into three subsets, with each subset exhibiting a different data distribution pattern serving distinct learning tasks, as depicted in Figure 6.

The main subset, referred to as subset A, which encompasses the most extensive conformation data, contains 20 million conformations from more than 110,000 molecules. Unlike previous datasets that only sample equilibrium conformations at local minima, our subset A consists of equilibrium conformations at local minima and near-equilibrium conformations additionally sampled around local minima. These near-equilibrium conformations aid in training models and reconstructing high-precision potential energy surfaces. Due to its more comprehensive conformation sampling method and broad distribution of heavy atoms, subset A can be used for various learning tasks, such as neural network potential (NNP) regression tasks (Kocer et al., 2022), machine learning force field (MLFF) tasks (Fu et al., 2023), or denoising-like pretraining tasks (Zaidi et al., 2023).

To introduce a higher level of complexity and challenge, we present the second subset, referred to as subset B, which includes 2.4 million conformers generated from approximately 1,400 molecules.

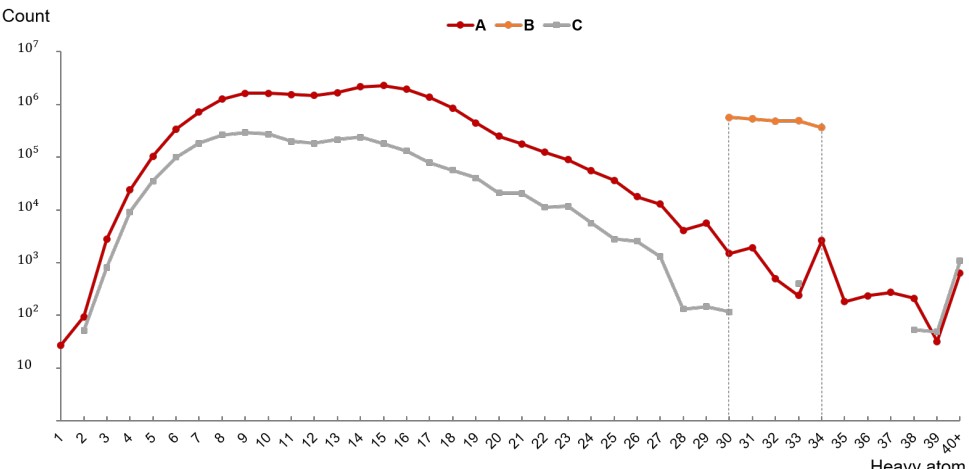

Figure 6: Distribution of the number of heavy atoms over sub-datasets

This subset consists of carefully selected representative drug molecules, based on domain expert annotations, with a large number of heavy atoms ranging from 30 to 34, as shown in Figure 6. This subset facilitates multiple tasks, such as testing the model's extrapolative and generalization capabilities and assessing its performance in real drug design workflows.

The third part, referred to as subset C, includes molecules that are non-analogous to those in subsets A and B. Subset C can be used for potential-related tasks either as a supplementary data source combined with the training set or as a validation set. Since the three subsets contain molecules that occupy distinct and separate regions in the chemical representation space, researchers have the flexibility to combine them in various ways.

### 4.5 COMPUTATION COST

All data preparation and DFT calculations were performed on a high-performance computing (HPC) cluster. In total, the computations utilized approximately 10 million core-hours of CPU resources.

## 5 BENCHMARK RESULTS

Potential energy prediction is one of the most important benchmark tasks in the field of computational chemistry, as it serves as the foundation for numerous downstream tasks such as reaction simulations (Manzhos & Carrington, 2021), protein dynamics (Majewski et al., 2023), and crystal structure screening (Chen & Ong, 2022). Additionally, the potential energy prediction task is typically employed to evaluate whether the model has successfully learned robust representations of molecular geometries (Gasteiger et al., 2020b;a; Liao & Smidt, 2023; Liu et al., 2024). Potential energy prediction task leverages the 3D structure of molecules as input to predict the potential energy of each conformation. In this section, we will discuss the results of benchmark models on the potential energy prediction task using the QO2Mol dataset.

### 5.1 DATA PREPROCESS PIPELINE

It has been successfully demonstrated that utilizing predefined atomic reference energies to optimize the model's prediction target enables the neural network to focus on fitting conformational energies. This approach can be represented by the following formula:

$$E_f = E_m - \sum_e N_e \epsilon_e \qquad (1)$$

where $E_f$ denotes formation energy, $E_m$ denotes molecule energy. $N_e$ corresponds to the number of atoms of element $e$ and $\epsilon_e$ corresponds to the reference energy of single atom of element $e$. Such

strategy has been demonstrated to effectively reduce the variance in energy fitting, enhancing the stability of training and the performance of the model on large-scale dataset. Notably, the top-ranked teams in the CFFF Prize all employed this approach.

## 5.2 BENCHMARK MODELS

In this section, we mainly consider two types of benchmark models: invariant models and equivariant models. Invariant models, such as SchNet (Schütt et al., 2017), SphereNet (Zhao et al., 2023), DimeNet++ (Gasteiger et al., 2020a), GemNet (Gasteiger et al., 2021), leverage features that remain unchanged under rotations and translations. These features include interatomic distances, bond angles, and torsion angles. By focusing on invariant features, these models can effectively capture the essential geometric relationships within molecular structures without being affected by their spatial orientation. Equivariant models or approximately Equivariant model, such as Equiformer (Liao & Smidt, 2023), EquiformerV2 (Liao et al., 2024), and eSCN (Passaro & Zitnick, 2023), utilize features that transform predictably under rotations and translations. These features include the irreducible representations of the SO(3) group and higher-order interactions. Equivariant models are designed to handle the inherent symmetries of molecular systems, allowing them to better capture the directional dependencies and interactions between atoms. Notably, most of these benchmark models were adopted by participants in the CFFF Prize. By employing both invariant and equivariant models as benchmarks, we can comprehensively evaluate the performance and robustness of various approaches in capturing the complexities of molecular structures and dynamics.

## 5.3 POTENTIAL PREDICTION BENCHMARK

We first evaluate the interpolation performance of potential prediction task over a series of benchmark models on subset A , which is aforementioned in Section 4.4. Subsequently, we undertook a more challenging task of employing these trained models to predict potential energies on the subset B, in order to evaluate the extrapolation capability of benchmark models. The results are presented in Table 3. We employ Mean Absolute Error (MAE) as the evaluation metric, measured in units of kcal$\mathring{u}$mol$^{-1}$. Detailed experimental settings are provided in the Appendix D.

Table 3 presents that GemNet stands out with the lowest MAE on test set A and relatively high generalization capability on test set B, indicating exceptional performance with a moderate number of parameters. Spherenet and SchNet, show higher MAE, reflecting limited expressive power. Equiformer and eSCN demonstrate good performance with lower MAE, balancing parameter count and accuracy effectively.

Table 3: MAE results on potential prediction task in units of kcal$\mathring{u}$mol$^{-1}$.

| Model | Params | Interpolation | Extrapolation |
|---|---|---|---|
| Spherenet | 2.7M | 0.10522 | 3.29613 |
| Equiformer | 3.5M | 0.07743 | 2.22257 |
| DimeNet++ | 5.0M | 0.07681 | 4.40856 |
| SchNet | 5.7M | 0.12974 | 8.73877 |
| GemNet | 5.7M | 0.02357 | 2.85464 |
| eSCN | 17.1M | 0.06417 | 3.60763 |
| EquiformerV2 | 38.0M | 0.04757 | 2.88512 |

## 6 CONCLUSION

In this paper, we present the QO2Mol database, a open-source large-scale data resource designed for organic molecular researchs. This database comprises 120,000 organic molecules generated from real compound libraries. The collection includes more than 20 million conformers, reflecting significant structural diversity and complexity. With representation from 10 different elements and heavy atom counts exceeding 40, the QO2Mol database offers an extensive and diverse molecular landscape for research exploration.Despite the richness and diversity of the dataset, it may not cover all possible molecular configurations or adequately represent certain chemical environments. Future research endeavors could involve leveraging the diverse and extensive molecular data within the QO2Mol database to refine and optimize machine learning applications in the field of computational chemistry.

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

# A    KEY INFORMATION

**Dataset documentation**    All the documentation for our datasets, along with usage demo scripts via Python, are provided at `https://github.com/ikovey/QO2Mol`.

**Author statement**    We bear all responsibility in case of violation of rights, etc., and confirmation of the data license.

**License**    This work uses **CC BY-NC-SA 4.0**. See details at `https://creativecommons.org/licenses/by-nc-sa/4.0/`.

**Maintaining Plan**    We utilize persistent cloud storage servers to provide accessing and downloading of the dataset. Further version will be updated upon research demands and the latest available links will be provided on the official Github repository.

# B    DATA FILE FORMAT

The QO2Mol database comprises several chunk files, each containing a list of molecular data objects. The description of the fields in each molecule object is provided in Table S1. We also provide a supplementary bunch of thermochemical properties at local minima to facilitate further research, with field names depicted in Table S2. Given the same data formats across all sets, researchers retain the flexibility to conduct data preprocessing or resplitting utilizing alternative methodologies.

Table S1: Data File Format

| Field | Description |
|---|---|
| inchikey | String, the identity of the conformer. |
| confid | String, the identity of the conformer. |
| atom_count | Integer, the number of atoms in the molecule. |
| bond_count | Integer, the number of bonds in the molecule. |
| elements | List, length equal to the number of atoms. Each value indicates the atomic number in the periodic table. |
| coordinates | List, length equal to the number of atoms. Each element is a 3-tuple representing the 3D coordinates (x, y, z) of the corresponding atom. |
| edge_list | List, length equal to the number of bonds multiplied by 2. Each element (i, j) represents an edge from atom i to atom j. |
| edge_attr | List, length equal to the number of bonds multiplied by 2. Each value represents a bond type. '1': single bond, '2': double bond, '3': triple bond. |
| energy | Float, the calculated potential energy of the molecule. |
| force | List, length equal to the number of atoms multiplied by 3. Each element represents the force component (x, y, z) of an atom. |
| net_charge | Float, the overall charge of a molecule. |
| formal_charge | List, length equal to the number of atoms. Each element represents the formal charge of the corresponding atom. |

# C    CHEMICAL SPACE

Relative to the QM9 database, which is limited to the elements C, H, O, N, and F, QO2Mol dataset encompasses a broader range of elements commonly found in organic molecules. These include C, H, O, N, S, P, F, Cl, Br, and I, which depicts the number of molecules in our dataset and QM9 containing for each element. QO2Mol dataset comprises a signiffcantly larger number of molecules that contain the element F, totaling 10,345 compounds, in contrast to the mere 310 F-containing molecules in QM9. Additionally, our dataset includes a substantial number of molecules containing S (29,702), P (2,464), Cl (9,829), Br (2,549), and I (647) elements, all of which are absent from

Table S2: Supplementary Thermochemical Properties

| Field | Description |
|---|---|
| inchikey | String, the identity of the conformer. |
| confid | String, the identity of the conformer. |
| dipole | List, length equals 3 corresponding to Cartesian coordinate components. |
| esp_charge | List, length equals number of atoms. |
| mulliken_charge | List, length equals number of atoms. |
| freq | List, length equals 3N-6. N denotes number of atoms. |
| hessian | List, the upper triangular version of hessian matrix. Length equals 3N(3N+1)/2. |
| thermochem | Dict, containing 7 items: capacity, enthalpy, entropy, free_energy, thermal_e, total_e. |

the QM9 database. This expanded elemental coverage in our dataset enables a more comprehensive exploration of the chemical space, encompassing a wider array of important and diverse molecular structures.

Table S3 summarizes the presence of ring structures in the molecules. Rings are essential components of organic molecules, and the majority of drug molecules contain ring structures. Due to the influence of ring strain, 5-membered and 6-membered rings are more stable compared to 3-membered and 4-membered rings. It is evident from the results of the QO2Mol databases that molecules containing 5-membered and 6-membered rings are more prevalent. However, due to the limitations on heavy atom counts, the QM9 database includes a greater number of molecules with 3-membered and 4-membered rings. Aromatic rings represent a distinct category of ring structures, contrasting with aliphatic rings. Aromatic rings can be 5-membered, such as pyrrole and furan, or 6-membered, such as benzene and pyridine. Due to their high stability, aromatic rings are commonly encountered in organic molecules. In the ChEMBL library, the majority of molecules contain aromatic rings, and a significant proportion of molecules in the QO2Mol database also feature aromatic ring. However, the QM9 database exhibits a relatively lower percentage of molecules with aromatic rings, particularly 6-membered aromatic rings.

Table S3: Summary of the presence of ring structures in the molecules

|  |  | QO2Mol | QM9 |
|---|---|---|---|
| Ring Size | 3 | 3304 | 54489 |
|  | 4 | 3335 | 50720 |
|  | 5 | 53476 | 50951 |
|  | 6 | 72420 | 19527 |
|  | 7 | 4819 | 4465 |
|  | > 7 | 1453 | 750 |
| Ring property | Aromatic (5) | 28264 | 12209 |
|  | Aromatic (6) | 45645 | 3239 |
|  | Non-aromatic | 46094 | 114552 |

# D EXPERIMENT DETAILS

We conducted all experiment on A100 GPU cluster. For the interpolation task, we employ a 72%/18%/10% split for training, validation and testing on subset A. For the extrapolation task, we use the entire subset B. In our experiments, we established the basic parameter settings as follows. The cutoff radius is set to 5.0 angstrom for all models. The training process was conducted using the AdamW optimizer with a cosine annealing learning rate scheduler. For hyper-parameter optimization, we employed a grid search strategy. Target hyper-parameters include learning rate, batch size, and the weight decay, with the following ranges: learning rate {1e-3, 4e-4, 8e-4}, batch size {32, 64, 128, 256}, weight decay {0, 1e-5, 1e-4}. Each combination of hyper-parameters was evaluated on the valid set, and the configuration yielding the highest validation accuracy was

selected for the final model. Convenient data loading scripts and relative codes are available at https://github.com/ikovey/QO2Mol/.