# OpenReview forum: "An Open Quantum Chemistry Property Database of 120 Kilo Molecules with 20 Million Conformers"
_ICLR.cc/2025/Conference — Submitted to ICLR 2025_

### Official Review · Reviewer_KztH · 2024-10-24

**Soundness:** 2
**Presentation:** 1
**Contribution:** 1
**Rating:** 3
**Confidence:** 4

**Summary:**

This paper presents a benchmark dataset, named the Quantum Open Organic Molecular (QO2Mol) database, which contains 120,000 organic molecules and approximately 20 million conformers. The authors highlight that this dataset covers 10 different elements, (C, H, O, N, S, P, F, Cl, Br, I) with heavy atom counts exceeding 40. The quantum mechanical properties of the organic molecules were computed using the B3LYP/def2-SVP calculation method, which is claimed to provide accurate computations of properties such as potential energy and forces. The paper presents benchmark results for potential energy prediction, evaluating the performance of both invariant and equivariant models.

**Strengths:**

The strength of this paper may come from the large number of organic molecules included in the QO2Mol database and its diverse coverage of elements. Specifically, the database includes molecules with heavy atom counts exceeding 40, representing an improvement over existing datasets like QM9 and ANI-1. Another positive aspect is the inclusion of tasks related to conformer generation.

**Weaknesses:**

The most significant weakness of this paper is its fundamental lack of novelty when compared to the existing dataset in this chemistry field. The existing nabla^2DFT dataset [1] contains 2 million molecules, which is more than the 120,000 molecules in this QO2Mol database, and includes 16 million conformers, a number comparable to this work. The comparison becomes even more unfavorable when considering the depth of information provided. The nabla^2DFT dataset offers 17 different molecular properties, along with Hamiltonian and overlap matrices, and a wavefunction object - features that are not provided in this QO2Mol database. Additionally, the nabla^2DFT dataset includes relaxation trajectories for a significant number of organic molecules, another feature not present in this paper. Given that the nabla^2DFT dataset was made available online in June 2024, well before the ICLR submission deadline, it is unclear what distinct advantages the QO2Mol database offers over the existing nabla^2DFT dataset. Furthermore, this paper presents a significant lack of benchmark results, only providing potential energy prediction benchmarks in the main manuscript, which limits its scope and impact.

[1] Khrabrov, Kuzma, et al. "$\nabla^ 2$ DFT: A Universal Quantum Chemistry Dataset of Drug-Like Molecules and a Benchmark for Neural Network Potentials." arXiv preprint arXiv:2406.14347 (2024).

**Questions:**

Major Issues:
1. The authors need to explain the absence of citation to the nabla^2DFT dataset, which is a significant oversight given its relevance and prior availability.
2. The authors should clearly articulate the unique advantages of the QO2Mol database compared to the nabla^2DFT dataset
3. The limited scope of benchmark results requires further explanation. The focus solely on potential prediction experiments in the main manuscript seems insufficient for a benchmark dataset paper of this nature.
4. Although the authors emphasize the high precision of B3LYP, the choice still requires justification. B3LYP's performance may degrade with non-covalent and long-range interactions, which can be present in organic molecules. The authors should explain why B3LYP was chosen over ωB97X-D, which potentially offers better precision.

Minor Editorial Issues:
1. In the appendix section, there are typos such as 'inffuence' (should be 'influence') and 'signiffcant' (should be 'significant'), which diminishes the professionalism of the paper.
2. The spelling of ChEMBL is inconsistent throughout the paper, appearing as both "ChemBL" (appendix page 16) and "ChEMBL" (elsewhere). This inconsistency should be corrected for clarity.

---

### Official Review · Reviewer_w5GN · 2024-11-02

**Soundness:** 1
**Presentation:** 2
**Contribution:** 1
**Rating:** 3
**Confidence:** 4

**Summary:**

The paper proposes a new dataset and benchmark with quantum chemical (QC) properties of 120,000 organic molecules and over 20 million conformers. The dataset uses moderately accurate physical theory B3LYP/def2-SVP. Moreover, the paper demonstrates the performance of 7 modern Neural Network Potentials on the proposed benchmark.

**Strengths:**

- A new dataset with QC properties and a new scheme for conformer generation.
- A new benchmark for NNPs.

**Weaknesses:**

- Most recent datasets with the same or higher level of theory are not cited and compared with, for example, QMugs [1], SPICE [2], nablaDFT [3], and QM1B [4]. Moreover, the cited PubchemQC [5] dataset contains 86 million structures, while PCQM4Mv2 [6] is its subset.
- The comparison between theory levels is unclear: for example, the level of theory in ANI-1 has more complex exchange correlation potential than in other listed datasets.
- The conformer generation procedure seems undermotivated. Why were 30-degree increments for the angles chosen instead of random or more physically based?
- The dataset composition scheme that involves fragmentation may not be optimal since the molecules that are included in the dataset are the most frequent fragments of bigger molecules. Such dataset may not cover the compound space well enough as it does not contain molecules that are not fragments of larger molecules. This may explain the performance drop between the subset A test set and the subset B.
- Finally, it is not clear how the subset B was generated.  Were these molecules chosen from the initial dataset or taken from another source? Atom and scaffold analysis may reveal severe divergence between A and B.

**Questions:**

- Listed in Cons.
- Provide a comparison with other datasets.
- Provide a detailed comparison between subsets A and B.
- Add the description of the molecular fragmentation method.
- Typo in line 185: We provides.

[1] Isert, C., Atz, K., Jiménez-Luna, J., & Schneider, G. (2022). QMugs, quantum mechanical properties of drug-like molecules. Scientific Data, 9(1), 273.
[2] Eastman, P., Pritchard, B. P., Chodera, J. D., & Markland, T. E. (2024). Nutmeg and SPICE: Models and data for biomolecular machine learning. Journal of Chemical Theory and Computation.
[3] Khrabrov, K., Ber, A., Tsypin, A., Ushenin, K., Rumiantsev, E., Telepov, A., ... & Kadurin, A. (2024). ∇ DFT: A Universal Quantum Chemistry Dataset of Drug-Like Molecules and a Benchmark for Neural Network Potentials. CoRR.
[4] Mathiasen, A., Helal, H., Klaser, K., Balanca, P., Dean, J., Luschi, C., ... & Masters, D. (2023). Generating QM1B with PySCF $ _ {\text {IPU}} $. Advances in Neural Information Processing Systems, 36, 55036-55050.
[5] Nakata, M., Shimazaki, T., Hashimoto, M., & Maeda, T. (2020). PubChemQC PM6: Data sets of 221 million molecules with optimized molecular geometries and electronic properties. Journal of Chemical Information and Modeling, 60(12), 5891-5899.
[6] Hu, W., Fey, M., Ren, H., Nakata, M., Dong, Y., & Leskovec, J. OGB-LSC: A Large-Scale Challenge for Machine Learning on Graphs. In Thirty-fifth Conference on Neural Information Processing Systems Datasets and Benchmarks Track (Round 2).

---

### Official Review · Reviewer_vr1N · 2024-11-02

**Soundness:** 2
**Presentation:** 1
**Contribution:** 2
**Rating:** 3
**Confidence:** 4

**Summary:**

This paper introduces a computational dataset of small to medium-sized molecules. For 120k molecules, energies and atomic forces were computed for ~170 conformers per molecule. The molecules were generated through fragmentation and recombination of drug-like compounds in the CHEMBL library. The process of database construction (molecule library enumeration, conformer generation, electronic structure calculations) is described. Eventually, the study includes a comparison of established molecular machine learning models applied to in-domain and out-of-domain prediction tasks within this dataset.

**Strengths:**

1. High-level reference data on molecular energies and forces are necessary for creating more generalizable ML potential energy surfaces and force fields, with applications e.g. in drug discovery.
2. The fragmentation–recombination strategy is chemically sensible, ensuring that the included molecules are realistic and feasible.

**Weaknesses:**

The main weaknesses of the paper are significant misinterpretations of core chemistry concepts and a lack of clarity around the dataset’s intended applications, leading to technical ambiguities and misleading claims.

1. The paper does not clearly articulate the goal of developing the dataset, or potential use cases. Although the data is well-suited for potential energy surface modeling, the introduction implies applications outside of this scope (e.g., structure–activity relationships), which is strongly misleading. Section 2.2 should be re-written to clarify these use cases, and clarify the unclear “conformer generation” task. Moreover, the paper would strongly benefit from repositioning the use cases earlier in the paper to align the introduction with actual practical applications. A number of vague statements regarding applicability should be removed.

2. The discussion (and application) of quantum chemistry methodologies is severely flawed.
  * The high-level introduction to DFT misses a number of key points: The main goal is to approximate solutions to the electronic Schrödinger equation for obtaining energies and, possibly, estimate further molecular property labels from the computed solution.
  * Figure 2 and the accompanying discussion makes strong statements about computation accuracy using different basis sets. However, both the Pople basis sets (6-31G(d) / 6-31G(2df,p)) and the Karlsruhe basis sets (def2-SVP) are double-zeta basis sets with additional polarization functions. To the best of my knowledge, for the molecules studied, the accuracy differences between these basis sets should be negligible. Any claims of significant accuracy differences must be rigorously supported with citations.
  * Statements like “B3LYP/def2-SVP as one of the highest precision levels achievable“ are strongly misleading. In fact, in computational chemistry, double-zeta basis sets are considered a rather low-level method. Established workflows, such as optimizing geometries with cheap semi-empirical methods, followed by single-point DFT for energies and gradients, are entirely overlooked in generating the dataset.
3. The discussion of chemistry concepts contains a number of inaccuracies and misinterpretations which weaken the perceived credibility of the study. Examples (non-comprehensive selection) include:
  * Force fields and potential energy surface are often discussed as two disjoint entities – rather than explaining that force fields are formally the derivative of the potential energy surface.
* SMILES and InChI representations are introduced without clarity on their primary function as molecular graph notations.
* “Heavy atom is any atom other than hydrogen, typically used in molecular studies to focus on more complex atomic interactions” (p. 3) This statement fails to address why distinguishing heavy atoms is relevant to real-world applications.
* “Labeled as important phosphate groups by our chemistry expert” (p. 5) What is the rationale behind labeling phosphate groups (which are one specific out of many relevant functional groups)?
* “while the significance of other conformations is also noteworthy” (p. 6) This statement is highly ambiguous. What is the relevance of further (non-minimum) conformations?

4. The choice of MAE as the error metric for model evaluation overlooks an essential consideration. Since energy scales with molecular size (i.e., number of heavy atoms), the MAE will naturally increase in an out-of-domain prediction task that includes larger molecules. The (in)ability to generalize across chemical space is not reflected in this metric.

5. The paper uses overly promotional language, particularly in the abstract and introduction. Examples include “unprecedented innovation and efficiency”, “professional and transformative research”, “high-precision […] quantum mechanical level”, “meticulously computed”, “comprehensive studies of structure–property relationships”. These sections should be thoroughly revised.

**Questions:**

1. The paper provides insufficient detail on the fragmentation / recombination strategy to construct the molecule library. What constitutes a fragment? How were the fragments generated and selected?
2. Similarly, the rationales and workflows of conformation generation and selection of “stable” conformations should be explained in more detail (at minimum, in the Appendix).
3. A notable portion of the paper blend descriptions of the dataset and comparisons to existing datasets, with overlapping content in section 2.2 and section 3. A clearer separation of these aspects would improve clarity and aid reader comprehension.
4. A better distinction between *molecules* and *conformations* would be beneficial for readability of the paper, and for understanding the structure of the dataset. E.g. the analyses in Fig. 5a and Fig. 6 would be more informative on the molecule level (rather than on the conformation level). A plot of average conformation number against heavy atom count, for example, would help contextualize the dataset structure. Similarly, plots of further simple-to-compute topological molecular descriptors (molecular weight, ring counts, …) would add further information.
5. The purpose of Fig. 5b, showing energy as a function of dihedral angle without specifying the molecule(s) is unclear. From my perspective, this plot, as it stands, is elementary, and provides no insights into the dataset.
6. An estimate of the computational resources required for dataset generation (in terms of CPU hours), would provide readers with a realistic perspective on the cost, and feasibility of expanding such datasets.

---

> ### Author Response · Authors · 2024-11-18
> **Response Part1**
>
> Dear Reviewe rvr1N:
>
>
> Highly constructive feedback we would like to express our deep gratitude to！
> Since your feedback is very detailed, please don't mind if we address it section by section.
>
>
> # 1) The goal and potential application scenario
>
> - The paper does not clearly articulate the goal of developing the dataset, or potential use cases. Section 2.2 should be re-written to clarify these use cases, and clarify the unclear “conformer generation” task.
>   - We have revised the paragraph describing the potential application scenarios in introduction in the revised version;
>   - We have clarified the task of conformation generation with more precise language. Additionally, we have included relevant references in the revised version;
>
> # 2) Basic concepts introduction
>
> - Figure 2 and the accompanying discussion makes strong statements about computation accuracy using different basis sets.
>   - In fact, the original figure simply illustrated the classification of different methods and basis sets. To avoid any misleading, we have removed Figure 2 and adjusted the statements related to accuracy.
> - Force fields and potential energy surface are often discussed as two disjoint entities – rather than explaining that force fields are formally the derivative of the potential energy surface.
>   - We have revised the relevant text to better align with reasonableness.
>
> - The high-level introduction to DFT misses a number of key points: The main goal is to approximate solutions to the electronic Schrödinger equation for obtaining energies and, possibly, estimate further molecular property labels from the computed solution.
>   - We have added this point to the main text in the revised version.
> - SMILES and InChI representations are introduced without clarity on their primary function as molecular graph notations.
>   - We have added their encoding method of molecular graphs in the revised version.
> - “Heavy atom is any atom other than hydrogen, typically used in molecular studies to focus on more complex atomic interactions” (p. 3) This statement fails to address why distinguishing heavy atoms is relevant to real-world applications.
>   - We have added why distinguishing heavy atoms is important in the revised version.
> - Statements like “B3LYP/def2-SVP as one of the highest precision levels achievable“ are strongly misleading. In fact, in computational chemistry, double-zeta basis sets are considered a rather low-level method.
>
>   - Actually, the complete sentence of the original text is "one of the highest precision levels **achievable within an acceptable computational cost range** for **large-scale calculations** of organic molecular systems. "
>   - We believe this description is accurate. *B3LYP* is indeed one of the highest accuracy choices widely used in practice for datasets **with tens of millions of samples**.

---

> ### Author Response · Authors · 2024-11-18
> **Response Part2**
>
> # 3) Data generation
>
> - Established workflows, such as optimizing geometries with cheap semi-empirical methods, followed by single-point DFT for energies and gradients, are entirely overlooked in generating the dataset.
>
>   - We point out that the workflow we employed as detailed in Section 4 is optimizing geometries with  *B3LYP/def2-SVP*, followed by single-point calculation with  *B3LYP/def2-SVP*. Our workflow is more accurate than optimizing structures with cheap semi-empirical methods.
> - “Labeled as important phosphate groups by our chemistry expert” (p. 5) What is the rationale behind labeling phosphate groups (which are one specific out of many relevant functional groups)?
>   - By labeling some functional groups that play important roles in molecules, we can prevent accidental deletion during the screening process. For example, phosphate groups often influence various properties of molecules.
> - “while the significance of other conformations is also noteworthy” (p. 6) This statement is highly ambiguous. What is the relevance of further (non-minimum) conformations?
>   - That's a great point! One of the biggest features of our dataset is the relatively high sampling density of conformations for the same molecule. On average, there are dozens of conformations for each molecule in our dataset, while other datasets typically have only a few structures per molecule (usually only local minima are included). It is well known that neural network potentials are reliable within the data region but exhibit instability outside of the sampling area. Therefore, expanding the sampling area can help achieve a larger confidence interval and build a more stable model. Consequently, conformations in non-equilibrium regions are also important. Especially when you want to do some long-time MD experiments.
> - The paper provides insufficient detail on the fragmentation / recombination strategy to construct the molecule library. What constitutes a fragment? How were the fragments generated and selected?
>   - The fragment algorithm is not the main contribution of this work and, therefore, has not been elaborated upon in detail here. A comprehensive discussion of the fragment algorithm will be presented in an independent publication later. For simplicity, we employed a technique similar to [1], where a set of rules is defined to retain meaningful chemical substructures and recombine chemical motifs.
> - An estimate of the computational resources required for dataset generation (in terms of CPU hours), would provide readers with a realistic perspective on the cost, and feasibility of expanding such datasets.
>   - We added a new subsection in Section 4 to describe the computation cost.
>
> [1] J. Degen, C. Wegscheid‐Gerlach, A. Zaliani, and M. Rarey, “On the art of compiling and using ‘drug‐like’ chemical fragment spaces,” *ChemMedChem*, vol. 3, no. 10, pp. 1503–1507, Oct. 2008, doi: [10.1002/cmdc.200800178](https://doi.org/10.1002/cmdc.200800178).

---

> ### Author Response · Authors · 2024-11-18
> **Response Part3**
>
> # 4) Writing and Figures
>
> - The paper would strongly benefit from repositioning the use cases earlier in the paper to align the introduction with actual practical applications. A number of vague statements regarding applicability should be removed.
>   - We have moved the relevant content regarding potential application scenarios to the beginning of Section 2. And deleted some of the statements.
> - The paper uses overly promotional language, particularly in the abstract and introduction. Examples include “unprecedented innovation and efficiency”, “professional and transformative research”, “high-precision […] quantum mechanical level”, “meticulously computed”, “comprehensive studies of structure–property relationships”. These sections should be thoroughly revised.
>   - We have adjusted some adjectives and adverbs and removed certain phrases to make the descriptions more precise.
>
> - A better distinction between *molecules* and *conformations* would be beneficial for readability of the paper, and for understanding the structure of the dataset. E.g. the analyses in Fig. 5a and Fig. 6 would be more informative on the molecule level (rather than on the conformation level). A plot of **average conformation number against heavy atom count**, for example, would help contextualize the dataset structure. Similarly, plots of further simple-to-compute **topological molecular descriptors** (molecular weight, ring counts, …) would add further information.
>   - We have added a table with respect to **average conformation number** and other **molecular descriptors** in the main text.
>   - We believe the heavy atom numbers are also informative for some tasks (e.g. transfer learning tasks) and so we keep the figure.
> - The purpose of Fig. 5b, showing energy as a function of dihedral angle without specifying the molecule(s) is unclear. From my perspective, this plot, as it stands, is elementary, and provides no insights into the dataset.
>   - Thank you for your feedback. Our intention of Fig. 5b was to convey the comprehensiveness of our sampling method, meaning that the entire space around a flexible bond is evenly scanned in nearly all 360 degrees. And we did the same rotational scan to every flexible bond of every molecule in the dataset. It's possible that our statement was not sufficient to convey this point. We will revise some of the sentences.
>
> ---
> Once again, we want to raise our sincere gratitude to you!
>
> We have absorbed many of your professional and detailed suggestions.
>
> These valuable insights have truly enhanced the quality of the paper. Many Thanks!

---

### Official Review · Reviewer_MtoR · 2024-11-04

**Soundness:** 2
**Presentation:** 1
**Contribution:** 1
**Rating:** 1
**Confidence:** 4

**Summary:**

QO2Mol, an open-source quantum chemistry dataset, is presented for organic molecular science research. It contains 120,000 molecules and 20 million conformers (10 elements, >40 heavy atoms) with potential energy and forces calculated at the B3LYP/def2-SVP level. Derived from ChEMBL fragments, the dataset is divided into three subsets: A (20 million conformations, >110,000 molecules, equilibrium and near-equilibrium), B (2.4 million conformers, ~1,400 drug molecules, numerous heavy atoms), and C (dissimilar molecules). Benchmark results for potential energy prediction using various models are included. The dataset and code are available on GitHub.

**Strengths:**

This paper demonstrates several strengths. It addresses a recognized need for a large and diverse quantum chemistry dataset for training and evaluating machine learning models in molecular sciences. The inclusion of benchmark results using established models provides a baseline for future research. The open-source availability of the dataset and code further encourages community engagement and broader application.

However, the paper's strong focus on quantum chemistry methodologies and detailed discussions of molecular properties, conformations, and potential energy surfaces might be more suitable for a computational chemistry or materials science audience. While leveraging machine learning for benchmarking, the core contribution and primary impact lie within the domain of chemistry, potentially making it less central to the interests of a machine learning conference focused on algorithmic advancements or theoretical foundations of AI.

**Weaknesses:**

This manuscript and the accompanying database should be submitted to a chemistry journal. The work here is predominantly chemistry focused, primarily quantum physical chemistry. Details of DFT calcuations, the construction of dataset, and the utility of the dataset requires peer-review from experts in quantum chemistry. To highlight this, ANI-1 and QM9 are both published in Scientific Data. PCQM4v2 is published in NeurIPS, however the work is part of a larger body of benchmarks for OGB-LSC, and the DFT datasets were curated from a peer-reviewed chemistry work by Nakata et al. (2017) in Journal of Chemical Information and Modeling.

The paper provides a lot of details about how the dataset was assembled, but it is unlikely that the anonymous reviewers of this venue would be able to give proper feedback on this; it would be irresponsible to publish this work without proper peer-review. Only the final section 5.3, with Table 2, are there relevant machine learning benchmarking results on some deep models on the dataset. There is also a lot of importance placed on the size of the dataset (Figures 1, 2, 3), but it is not clear whether the additional conformations/fragmentation methods are physically sound or useful for chemical modelling.

Additional issues:
- Table 2 should have caption before the table.
- Figure 6 caption should be below the figure.
- There are gramatical errors that need to be addressed: for example, line 359, "The main subset, referred to as subset A, encompassing the most extensive conformation data, containing 20 million conformations from more than 110,000 molecules."
- A double period in line 350.

Additionally there are other works, not mentioned, that have explored something similar; with conformations, higher precision calculations, property annotations etc. (also published in chemistry related venues): GEOM set by Axelrod et al. (2022), QCDGE by Zhu et al. (2024), and Nabla2DFT and NablaDFT by Khrabrov et al. (2024), VQM24 by Khan et al. (2024).

**Questions:**

For example, Figure 2, the precision levels seem to be some linear scale of various basis sets. While I am not an expert in computation chemistry, I believe this figure paints a deceptively simple measure of precision in DFT. Are the choice of level of theory and basis sets not dependent on the system being looked at; and are they appropriately chosen? Are there any dispersion corrections (as done in GEOM, and QCDGE)? Is the choice of DFT level of theory appropriate for properties that are used as targets (ie. entropy, enthalpy, heat capacity, etc.)?

Is the fragmentation method deployed to generate the structures common used? There are no details given on the fragmentation algorithm used. The authors claim that this approach would give "a diverse range of significant and complex chemical space," with no evidence of this.

The conformation generation through cheminformatics software like RDKit seems low-precision, given the computational cost required for DFT at B3LYP/def2-SVP. GEOM for example uses CREST, which is based on semi-empirical tight-binding model. Is RDKit sufficient? What are convergence criteria typically used for optimization process of these structures?

Note that most of these questions would likely not require an explanation if reviewed by an expert in quantum computational chemistry.

---

> ### Author Response · Authors · 2024-11-16
> **Response Part1**
>
> Dear Reviewer MtoR,
>
> We sincerely appreciate the time and effort you have invested in reviewing our paper.
> We have carefully considered each of your comments and  below we provide a point-by-point response to your feedback.
>
> # 1) regarding the appropriateness of the submission topic
>
> We would like to first address the key concern you have raised.
>
> It is important to emphasize that **the name of this track is "Datasets and Benchmarks".** As is widely recognized in the field of deep learning, the quality of data sources is a critical issue. Our work is specifically designed to contribute to the development of higher-quality AI resources for the community. This dataset is tailored for training AI models, intended for use with AI models, and designed for fair comparisons of the performance of AI models. **These characteristics clearly align with the objectives of this track.**
>
> We kindly ask the reviewers to adopt a more inclusive perspective toward the concept of AI datasets.  AI-related fields such as organic molecular and protein design are also advancing rapidly recent years. The well-known AlphaFold has recently been awarded a Nobel Prize.
>
> Thus, **we kindly request the reviewers' understanding once again**, molecular datasets are an integral part of AI data resources, and **the concept of AI datasets should not be disriminatively limited to only "image datasets" and "text corpus datasets"**.
>
>
>
> # 2) regarding the main content about data generation
>
> Given that this manuscript is a dataset and benchmark paper, which is submitted to the "Datasets and Benchmarks" track, we believe it is our responsibility to provide a detailed explanation of the dataset generation process.
>
> In fact, many dataset papers fail to fulfill this obligation, resulting in a lack of transparency in the data production process, which creates significant challenges for users. A clear and explicit description of the production process is an essential feature of a high-quality benchmark dataset.
>
> Recently, AAAI 2025 hosted a "Good DATA" workshop, which elaborated on the issue of unclear data provenance in many fields. Perhaps visiting their [homepage](https://sites.google.com/servicenow.com/good-data-2025/) could help you understand the significant value of the detailed explanation we provided regarding our data production process.
>
> Furthermore, we would like to emphasize that **the evaluation criteria for data-focused papers should differ from those applied to model-focused papers**. The standards and writing conventions of model-focused papers should not be imposed on data-focused papers. We hope this perspective clarifies the unique considerations necessary for evaluating work in this category.

---

> ### Author Response · Authors · 2024-11-16
> **Response Part2**
>
> # 3) regarding the proposed relative works
>
>
> Below, we one by one discuss the 5 "potential related" works proposed by the reviewer.
>
> >[1] Khan, Danish, et al. "Towards comprehensive coverage of chemical space: Quantum mechanical properties of 836k constitutional and conformational closed shell neutral isomers consisting of HCNOFSiPSClBr." *arXiv preprint arXiv:2405.05961* (2024).
> >
> >[2]Zhu, Yifei, et al. "QCDGE database, Quantum Chemistry Database with Ground-and Excited-state Properties of 450 Kilo Molecules." *arXiv preprint arXiv:2406.02341* (2024).
> >
> >[3] Axelrod, S., Gómez-Bombarelli, R. GEOM, energy-annotated molecular conformations for property prediction and molecular generation. *Sci Data* **9**, 185 (2022). https://doi.org/10.1038/s41597-022-01288-4
> >
> >[4] Khrabrov, Kuzma, et al. "nabladft: Large-scale conformational energy and hamiltonian prediction benchmark and dataset." *Physical Chemistry Chemical Physics* 24.42 (2022): 25853-25863. https://doi.org/10.1039/D2CP03966D
> >
> >[5] Khrabrov, Kuzma, et al. "$\nabla^ 2$ DFT: A Universal Quantum Chemistry Dataset of Drug-Like Molecules and a Benchmark for Neural Network Potentials." *arXiv preprint arXiv:2406.14347* (2024).
>
>
>
>
>
> First of all, as you may have noticed, \[1\] \[2\] [5]  are not "published in chemistry related venues", and rather preprints that have not been peer-reviewed.
>
> As for \[3\], we quote **Line 215** in our paper: **Remark: We also acknowledge the existence of several other notable datasets in the field such as ...  these datasets focus on different domains and are not directly designed for the study of small organic molecules.**
>
> \[3\] are not designed for small organic molecules as you can read from its abstract that  **The Geometric Ensemble Of Molecules (GEOM) dataset contains ... data related to biophysics, physiology, and physical chemistry**.  And you can read from **Table3 in \[3\]** for more detailed information.
>
>
> As for [4], we list the main characteristics of the datasets.
>
> | dataset              | $\nabla$ DFT                                                 | $\nabla^2$ DFT                                               | QO2Mol                                                       |
> | :------------------- | :----------------------------------------------------------- | ------------------------------------------------------------ | :----------------------------------------------------------- |
> | Number of molecules  | 1 m                                                          | 1.9 m                                                        | 120 k                                                        |
> | Number of conformers | 5 m                                                          | 15 m                                                         | 20 m                                                         |
> | Maximum heavy atoms  | 27                                                           | 27                                                           | 44                                                           |
> | Elements             | $\mathrm{H}, \mathrm{C}, \mathrm{N}, \mathrm{O}, \mathrm{Cl}, \mathrm{F}, \mathrm{Br}$ | $\mathrm{H}, \mathrm{C}, \mathrm{N}, \mathrm{O}, \mathrm{Cl}, \mathrm{F}, \mathrm{Br}$ | $\mathrm{H}, \mathrm{C}, \mathrm{N}, \mathrm{O}, \mathrm{F}, \mathrm{P}, \mathrm{S}, \mathrm{Cl}, \mathrm{Br}, \mathrm{I}$ |
>
>
>
> References [4] and [5] emphasize the simplicity and general applicability of their datasets. However, as shown in the table, the average number of conformations sampled per trajectory in [4] is only 5. This is insufficient for constructing a high-precision potential energy surface (PES).
>
> Perhaps you may not be familiar with this specific area, but as intuition would suggest, higher sampling density and a greater number of conformations lead to more accurate reconstructions of the potential energy surface\[6-7\]. From a practical perspective, an average of 5 samples is inadequate for this purpose.
>
>
>
> We will add \[3\] and \[4\] into the references and discuss the difference.
>
>
>
> [6]Sicong Ma, Cheng Shang, Zhi-Pan Liu. Heterogeneous catalysis from structure to activity via SSW-NN method. *J. Chem. Phys.* 7 August 2019; 151 (5): 050901. https://doi.org/10.1063/1.5113673
>
> [7] P.-L. Kang, C. Shang, and Z.-P. Liu. “Glucose to 5-Hydroxymethylfurfural: Origin of Site-Selectivity Resolved by Machine Learning Based Reaction Sampling”. *J. Am. Chem. Soc.*, vol. 141, no. 51, pp. 20525–20536, Dec. 2019.http://dx.doi.org/10.1021/jacs.9b11535

---

> ### Author Response · Authors · 2024-11-16
> **Response Part3**
>
> # 4) regarding  DFT level of theory problems
>
>
> - In Figure 2, the precision levels are just categories, not linear scaled.
>
> - Are there any dispersion corrections (as done in GEOM, and QCDGE)?
>
>   - Today, the most commonly used dispersion corrections are DFT/D3[1] and DFT/D4[2] coming from Professor Grimme, the honored academic luminary in this field.  We directly quote the original text from Page 2 in [1] by him: `it has been designed from the very beginning as a correction for common functionals such as B3LYP, PBE, or TPSS that may be not optimal for noncovalent interactions`. For simplicity, it **may not be the optimal choice** to employ dispersion correction on `B3LYP` level of theory. Given the very low cost of D3/D4 calculations, users are free to decide whether to apply it during the data preprocessing stage.
>
>
> - Are the DFT  level of theory and basis sets appropriately chosen?  Is the choice of DFT level of theory appropriate for properties that are used as targets?
>
>   - Yes.  **B3LYP**, the level of theory we choose, is one of the most commonly used for small organic molecules. We refer to a voting result upon the question `Which theory you used most in DFT calculating` from a famous Chinese community of computational chemistry conducted on year 2024,  which gather the votes of hundreds of researches , see Figure1 on http://sobereva.com/706 .  Or directly the [figure link](http://sobereva.com/images/706/1.png).
>
>
>
> [1] S. Grimme, J. Antony, S. Ehrlich, and H. Krieg, “A consistent and accurate *ab initio* parametrization of density functional dispersion correction (DFT-D) for the 94 elements H-Pu,” *The Journal of Chemical Physics*, vol. 132, no. 15, p. 154104, Apr. 2010, doi: [10.1063/1.3382344](https://doi.org/10.1063/1.3382344).
>
> [2] E. Caldeweyher *et al.*, “A generally applicable atomic-charge dependent London dispersion correction,” *The Journal of Chemical Physics*, vol. 150, no. 15, p. 154122, Apr. 2019, doi: [10.1063/1.5090222](https://doi.org/10.1063/1.5090222).
>
>
>
> # 5) regarding  Data Generation Questions
>
>
>
> - There are no details given on the fragmentation algorithm used. The authors claim that this approach would give "a diverse range of significant and complex chemical space," with no evidence of this.
>   - This paper focuses on datasets and benchmarks. The fragment algorithm is not the main contribution of this work and, therefore, has not been elaborated upon in detail here. A comprehensive discussion of the fragment algorithm will be presented in an independent publication. For simplicity, we employed a technique similar to [3], where a set of rules is defined to retain meaningful chemical substructures and recombine chemical motifs. In fact, we **have provided evidence** by detailed comparison in **Appendix C**, showcasing the diversity of our generated conformational space compared to QM9.
>
> - The conformation generation through cheminformatics software like RDKit seems low-precision, given the computational cost required for DFT at B3LYP/def2-SVP. GEOM for example uses CREST, which is based on semi-empirical tight-binding model. Is RDKit sufficient?
>   - Actually RDKit is only used for an initial structure generation, and subsequently we conduct a Conformation Search Process with  **B3LYP/def2-SVP**, which is more expensive than semi-empirical methods employed by CREST. Refer to **Line 301** and **Line 316** for more details.
>
> - What are convergence criteria typically used for optimization process of these structures?
>   - The four criteria detailed in **Line 304** are exactly convergence criteria typically used for optimization process, as is also employed in GaUssian series[4].
>
>
>
> [3] J. Degen, C. Wegscheid‐Gerlach, A. Zaliani, and M. Rarey, “On the art of compiling and using ‘drug‐like’ chemical fragment spaces,” *ChemMedChem*, vol. 3, no. 10, pp. 1503–1507, Oct. 2008, doi: [10.1002/cmdc.200800178](https://doi.org/10.1002/cmdc.200800178).
>
> [4] Frisch, M. J. et al. Gaussian~16 Revision C.01 (2016). Gaussian Inc. Wallingford CT.
>
>
>
>
>
> # 6) regarding writing and figure problems
>
>
>
> We sincerely thank you for your valuable feedback.
>
> These detailed errors may not be noticeable without a thorough reading.
>
> We truly appreciate the time and effort you have dedicated to carefully reviewing our manuscript !
>
> In the revised version of PDF, we will address the issues related to the title and grammatical errors as suggested.

---

> ### Comment · Reviewer_MtoR · 2024-11-22
>
> Thank you for the response, although the belligerent and condescending tone is not appreciated.
>
> **Appropriateness of submission**:
> - The track is "Dataset **and** Benchmarks", however the authors have neglected the benchmarking part. There is very little detail in machine learning in this paper, and more importantly, no insights into what this new dataset/benchmark probes in the off-the-shelf models they have implemented.
> - The well-known AlphaFold was published in Nature, another peer-reviewed scientific journal. Furthermore, AlphaFold was trained on CASP, which are carefully curated data with experimental X-ray crystallography and NMR spectroscopy. This is quite different from a dataset generated with some invented fragmentation/conformer generation method, followed by slightly improved DFT calculations.
>
> **Main content about data generation**:
> - I don't understand the authors reponse to this. This reviewer (and also the other reviewers) have pointed out the arbitrary method of data/conformer generation that the authors have chosen, which is precisely the problem with works that lack "transparency" or "clear and explicit description of the production process".
>
> > Recently, AAAI 2025 hosted a "Good DATA" workshop, which elaborated on the issue of unclear data provenance in many fields. Perhaps visiting their homepage could help you understand the significant value of the detailed explanation we provided regarding our data production process.
> - I do not understand why the authors have directed me to a conference workshop that has not happened yet. Perhaps visiting their [hompage](https://sites.google.com/servicenow.com/good-data-2025/) would help you understand that this conference will take place in March 2025, and that it is not a proper source for citation.
> - It is insufficient to generate such a domain-specific dataset (for a "data-focused paper") with such scant detail provided on data generation, and vague, jargony, and inaccurate statements (as pointed out by reivewer `vr1N`) about quantum chemistry.
>
> **Relative works**:
> > First of all, as you may have noticed [...] [3] are not designed for small organic molecules as you can read from its abstract that The Geometric Ensemble Of Molecules (GEOM) dataset contains ... data related to biophysics, physiology, and physical chemistry.
> - You may have noticed, that GEOM **is** focused on small organic molecules. The dataset includes molecules from QM9. The "biophysics, physiology, and physical chemistry datasets" are all small organic molecule datasets. I recommend reading the paper, rather than making conclusions from the abstract of a paper.
>
> > Perhaps you may not be familiar with this specific area, but as intuition would suggest, higher sampling density and a greater number of conformations lead to more accurate reconstructions of the potential energy surface[6-7].
> - Yes. I am unfamiliar in this specific area, so I will defer to the critical responses from reviewers `vr1N` and `KztH`. This statement is neither intuitive or trivial. This is a machine learning conference.
> - Every reviewer agrees that this work overlaps with existing work from nabla2. Nabla2 even provides benchmarking *tasks* based on their dataset, with specialized *splits*, and results of 10 different models on the task.
> - Thank you for adding the references to the related works.
>
> **Regarding DFT level of theory**
> - Figure 2, which the authors have removed, is not plotting the categories because there was an axis with ordinality. This also does not answer the question about the simplification of Jacob's ladder of DFT.
> - A citation to a blogpost with no peer-review or any publishing guidelines is unconvincing. Regardless, even if B3LYP is the most popular level of theory, this does not answer the question: is it appropriate for the properties that you are using it for?
>
> **Data generation**:
> - The provided evidence of the fragmentation method, which the authors have still refused to elaborate on in the main text, is compared to QM9, which is an enumeration of all structures of 9 heavy atoms. The structures are usually non-physical, very small, and, in the words of the authors: "severely limited in the number of molecular structures, making them grossly inadequate for training large-scale models." This comparison is unconvincing that the fragmentation method is successful.
> - According to your procedures, you **do not** perform conformer search with B3LYP/def2svp. The search is performed with low-level RDKit, and the structure is **optimized** with B3LYP/def2svp. Please do not make dishonest claims.
>
> ---
>
> The author responses are insufficient. It is my opinion that the authors are quantum chemistry experts, using domain-specific jargon in hopes that reviewers who are not domain experts, at a machine learning conference, would believe the work to be more innovative and interesting than it is.
>
> As such I will further lower my score to 1.

---

### Meta-Review · Area_Chair_WJiD · 2024-12-18

**Metareview:**

This submission to the datasets and benchmarks track introduces QO2Mol, a computational dataset of small to medium-sized molecules. Reviewers expressed concerns on the presentation and the significance of the presented work.

The authors may also need to seriously consider how the presented data may be organized and accompanied with machine learning methods that may help advance molecular forward property prediction or generative design research.

**Additional Comments On Reviewer Discussion:**

The authors provided rebuttal responses to some raised concerns. However, the reviewers still have serious concerns about the submission on the presentation, significance, and scope of the presented work.

---

### Decision · Program_Chairs · 2025-01-22

Reject